# Impacts of climate change on agro-climatic suitability of major food crops in Ghana

**Abel Chemura** *, **Bernhard Schauberger, Christoph Gornott**

Potsdam Institute for Climate Impact Research (PIK), Member of the Leibniz Association, Potsdam, Germany

* chemura@pik-potsdam.de

## Abstract

Climate change is projected to impact food production stability in many tropical countries through impacts on crop potential. However, without quantitative assessments of where, by how much and to what extent crop production is possible now and under future climatic conditions, efforts to design and implement adaptation strategies under Nationally Determined Contributions (NDCs) and National Action Plans (NAP) are unsystematic. In this study, we used extreme gradient boosting, a machine learning approach to model the current climatic suitability for maize, sorghum, cassava and groundnut in Ghana using yield data and agronomically important variables. We then used multi-model future climate projections for the 2050s and two greenhouse gas emissions scenarios (RCP 2.6 and RCP 8.5) to predict changes in the suitability range of these crops. We achieved a good model fit in determining suitability classes for all crops (AUC = 0.81–0.87). Precipitation-based factors are suggested as most important in determining crop suitability, though the importance is crop-specific. Under projected climatic conditions, optimal suitability areas will decrease for all crops except for groundnuts under RCP8.5 (no change: 0%), with greatest losses for maize (12% under RCP2.6 and 14% under RCP8.5). Under current climatic conditions, 18% of Ghana has optimal suitability for two crops, 2% for three crops with no area having optimal suitability for all the four crops. Under projected climatic conditions, areas with optimal suitability for two and three crops will decrease by 12% as areas having moderate and marginal conditions for multiple crops increase. We also found that although the distribution of multiple crop suitability is spatially distinct, cassava and groundnut will be more simultaneously suitable for the south while groundnut and sorghum will be more suitable for the northern parts of Ghana under projected climatic conditions.

## 1. Introduction

The agricultural sector of tropical countries is at great risk from the impacts of climate change. This is because of changes in weather patterns, which determine yields and crop production in these areas [1, 2]. Projections show that about a third of the world's population will be living in these countries by 2050, and the impact on agriculture is regarded as the most important and immediate danger of climate change to society [3, 4]. In Ghana, agriculture employs more

**Data Availability Statement:** Data used in the analysis is curated on Zenodo (https://zenodo.org/record/3669955/) and publicly available (DOI: 10.5281/zenodo.3669955).

**Funding:** This work was funded by the German Ministry for Economic Cooperation and

Development (BMZ) in close collaboration with the Deutsche Gesellschaft für Internationale Zusammenarbeit (GIZ) GmbH under the pilot study in Ghana as part of the project "AGRICA – Climate risk analyses for identifying and weighing adaptation strategies in sub-Saharan Africa." The funders had no role in study design, data collection and analysis, decision to publish, or preparation of the manuscript.

**Competing interests:** I have read the journal's policy and the authors of this manuscript have the following competing interests: A. Chemura is serving as an academic editor for PLoS One.

than half of the population directly and indirectly and is important for in contributing to food security, gross domestic production (GDP) and balance of payments [5]. Changing climatic conditions pose significant threat to the growth of the agricultural sector in Ghana because heavy reliance on rain fed production and drought vulnerability, especially as less than 2% of the agricultural area is under irrigation [6, 7].

In Ghana, the country's agriculture sector is dominated by smallholder family farms that are predominantly rain fed and thus climate sensitive [8, 9]. Among the leading agricultural food commodities by harvest area in the country are cassava, maize, groundnuts and sorghum [10]. An understanding of the risk of climate change to the agriculture sector in Ghana is required to build resilience. Among the impacts of climate change on agriculture in Ghana are unpredictable and variable rainfall, increasing temperatures, and longer dry periods. Some studies have observed delays in the onset of rain seasons in some regions [11]. Other studies have observed that the changes in n the onset of rainfall are beneficial for some crops while detrimental to other crops and yet all these crops are important for the food security basket [12]. Therefore, an integrated assessment that indicated the impacts of climate change on multiple crops is required to provide a comprehensive picture of the impacts. This because the smallholder farmers, who form the majority of farmers in Ghana, rarely produce individual crops [13–15].

Despite massive developments in agricultural production technology, weather and climate still play a significant role in influencing agricultural production in Africa and elsewhere [16–18]. In particular, under rain-fed conditions the production potential of a crop depends on the climatic conditions of an area. Therefore, each crop will thrive within a specific climatic envelope that can be enhanced by management–yet climate change will alter satisfaction of these requirements and subsequently the geography of crop suitability [19]. Thus, climate change adaptation measures such as agricultural intensification, crop diversification, improved crop varieties and other management strategies needed to stabilize or enhance food production now and under projected climatic conditions should operate within the natural production domains that determine crop suitability. Crop suitability is a measure of the climatic and other biophysical characteristics of an area to sustain a crop production cycle to meet current or expected targets [20, 21]. When combined with climate projections, suitability assessments are used to gauge shifts in crop potential under climate change [22, 23]. Since the results are spatially explicit, the suitability models identify the areas where adaptation measures are mostly required to avert the consequences of a predicted decline in climatic suitability of the crops.

Despite the potential of role of multiple crops in the food basket, limited attention has been paid on assessing the impacts of climate change on multiple crops to provide farmers with options for diversification or crop switching. There are no explicit indications of which crop combinations work where and with what individual or combined production outcomes. Impact studies have also mostly focused on individual crops with testing of adaptation measures following the same pattern. To the best of our knowledge, there are no publications assessing multiple crop suitability at national or local levels to guide farmers to select most suitable crops for their areas in building resilience. Such assessments are imperative for spatially explicit targeted adaptation planning and investment under Nationally Determined Contributions (NDCs) for Ghana. Furthermore, assessing agricultural potential is important in achieving many sustainable development goals (SDGs) such as reduction of poverty (SDG1), averting hunger (SDG2), enhancing good health and well-being (SDG3), responsible consumption and production (SDG12), reducing impacts of climate change (SDG13) and sustenance of life on land (SDG15) [24, 25].

In this study, we applied crop climatic suitability models to assess the impact of climate change agro-climatic suitability for cassava, groundnuts, sorghum and maize in Ghana. Maize,

sorghum, cassava and groundnuts are important staple crops planted on nearly three million hectares annually, which is ~83% of all the cropped area in Ghana [26]. Furthermore, diets in Ghana include combinations of maize or sorghum, cassava and groundnut in various proportions [27, 28], making it important that these crops are available also under climate change. There is a paucity of data on spatially explicit climate change impact assessments and limited analysis of impacts on multiple crops in same areas. Therefore, the aim of this study was to assess the impacts of projected climate change on four important food crops in Ghana by mid-century. Specifically, we intended to (i) identify the determinants of crop suitability in Ghana, (ii) identify climate change impacts on crop climatic suitability for individual and multiple crops with maize, sorghum, cassava and groundnuts as key food crops.

## 2. Methods

### 2.1 Crop production data

Data used in modelling the climatic suitability were obtained from the Ministry of Food and Agriculture (MoFA)'s statistics department, which is an independent government organization that is responsible for collecting and compiling official agricultural statistics in Ghana. Crop yields for maize, sorghum, cassava and groundnut are reported in metric tons per hectare (dry mass). This is the ratio of total production per year in a district divided by total land cultivated for that crop in that district for that year. These datasets are obtained from the agricultural extension officers in each district who carry out crop cutting experiments to estimate production and cropped area annually. Yield data used were from 2006 to 2016.

### 2.2 Defining crop suitability classes

The production data for each of the four crops was split into four groups (optimal, moderate, marginal and limited) using percentiles of the average yield between 2006 and 2016 (Fig 1A).

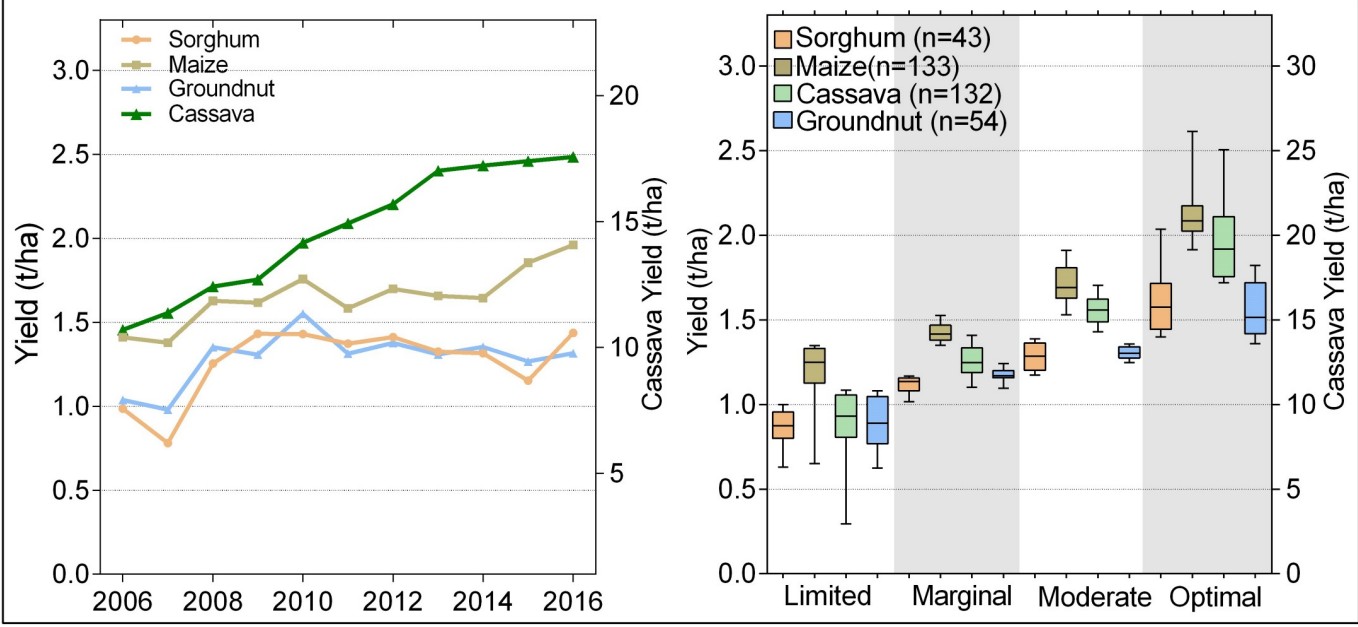

**Fig 1.** (a) Trends in yield for maize, sorghum, groundnut and cassava in Ghana from 2006 to 2016. (b) The 11 year mean crop yield distribution for each of the four suitability classes across all districts from measured data. The right axis in (a, b) is for cassava yields only.

Optimal suitable areas were defined as those areas that were above the 75th percentile of the mean yield of crop, representing areas with no significant limitations to sustained production and stability over time. Moderately suitable areas were those between the 50th and 75th percentile of the average yield, indicating areas with moderately severe limitations for sustained productivity or increased variability which increases the risk of crop failure. Marginally suitable areas had yield between the 25th and 50th percentile for the period, representing areas with severe limitations for sustained productivity, and pronounced variability between the years. Limited suitability areas are areas under the 25th percentile of the yield in the period, indicating areas where biophysical conditions are not apt for the crop, thus showing constantly low yields over time (Fig 1B). The assignment of one of the four groups (limited, marginal, moderate or optimal) was done per pixel and time frame. To ensure that data were in cropped areas, a NASA 2015 crop mask was used as obtained from the Global Croplands database [29].

## 2.3 Biophysical variables for crop suitability

This study applied an empirical supervised learning model to determine four crop suitability classes from agronomically important climatic and soil variables at national level [30, 31]. Eight biophysical parameters were used in modelling the climatic suitability of the four crops under current and future climatic conditions. These were total rainfall in the growing season, total rainfall received between March and September, sum of rainfall in the crop sowing month, rainfall coefficient of variation, diurnal temperature range between March and September, mean temperature growing season, mean temperature between March and September and top soil organic carbon (Table 1). The eight variables were selected because they are known to have major agronomic influence on the crops [32, 33]. The main growing period was defined as March to July in the South and end of May to September for the North according to distribution of rainfall and temperatures in Ghana [34] (S1 Fig). The precipitation variables were derived from the Climate Hazards Group InfraRed Precipitation with Station data (CHIRPS) daily data at 0.05 degrees resolution from 2006 to 2016 [35]. Temperature variables

**Table 1.  The eight biophysical variables used for crop suitability modelling and their descriptions and derivations from daily weather data.**

| Variable | Description* |
| --- | --- |
| Total rainfall in the growing season | Sum of rainfall for 24 May (DOY = 145) to 30 September (JD = 274) for the north and 1 March (JD = 61) to 30 June in the south. |
| Total rainfall received between March and September | Sum of rainfall received from 1 March (DOY = 61) to 30 September (DOY = 274) to represent the whole growing season for both north and south. |
| Sum of rainfall in the crop sowing month | Sum of rainfall received from 24 May to 30 June (DOY = 182) in the north and 1 March (DOY = 61) to 30 April (DOY = 121) in the south. |
| Rainfall coefficient of variation | The ratio of the standard deviation of the monthly sums of rainfall between March and September to the mean monthly rainfall. |
| Diurnal temperature range between March and September | The average of the differences between maximum temperature and minimum temperature from 1 March (DOY = 61) to 30 September (DOY = 274). |
| Mean temperature growing season | Average temperature between 24 May to 30 September (JD = 274) for the north and 1 March (DOY = 61) to 30 June (DOY = 182) in the south |
| Mean temperature between March and September | Average temperature between 1 March (DOY = 61) to 30 September (DOY = 274) to represent the whole growing season for both north and south |
| Top soil organic carbon | The amount of organic carbon in the top 5cm of the soil per ha. |

* DOY is Day of Year

were derived from the WFDEI Near Surface Temperature data for the same period at 0.5 degrees resolution [36]. Top soil organic carbon was obtained from ISIRIC [37].

For future climatic conditions, the same climatic variables used in model fitting were derived from data on projected climatic conditions for Ghana. We used climate projections of the Inter-Sectoral Impact Model Inter-comparison Project (ISIMIP) for the period 2006 to 2016 (baseline) and for 2041 to 2050 (future). This climate data consisted of four different general circulation models (GCM) projections, namely GFDL-ESM2M [38, 39], HadGEM-ES2 [40], IPSL-CM5A-LR [41], and MIROC-ESM-CHEM [42]. These GCMs were chosen because they are available with bias-adjustment [36, 43]. For future projections, the RCP2.6 and RCP8.5 scenarios were selected to represent the 1.5–2˚C-target of the Paris Agreement and a scenario without climate policy, respectively, to capture the range of climatic possibilities (Table 2). The modelling for climate impact assessment were run on the assumption of no change in soil organic carbon as there are currently no spatial near-future projections for this variable. All these variables were clipped to Ghana and then scaled, ensuring that they have a matching spatial resolution and extent. The same set of responses, predictors and scenarios were used for each crop and each scenario.

## 2.4 Modelling approach

Suitability models or their variants have been used in assessing the geography of crop suitability and in modelling impacts of climate change on agriculture for different crops. While the common approach is to use a 2 class (suitable/unsuitable) approach for modelling crop suitability [44–47], we propose a method that models four suitability classes (optimal, moderate, marginal and limited) as a 2 class system may over-estimate climate impacts by not scaling the suitability. Scaled four-class (high, moderate, marginal and unsuitable) suitability models are an alternative for determining suitability classes of agricultural crops from machine learning algorithms [31, 48–51]. To model the four suitability classes of the four crops, we applied the eXtreme Gradient Boosting (XGBoost) machine learning approach to the variables. XGBoost is an improvement of the recursive tree-based partitioning method of gradient boosting machines (GBM) by Friedman [52]. Gradient boosting is a technique implemented in a complex prediction model by iterative combinations of ensembles of weak prediction models into

**Table 2. Projected rainfall and temperature changes as used in the suitability modelling under the RCP2.6 and RCP8.5 for Ghana.** Variables are summarized across the country.

| Scenario | Model | Sum rainfall Mar -Sept(mm) | Sum rain sowing month (mm) | Rainfall coefficient of variation (%) | Rainfall growing season (mm) | Average Tmax —Tmin (T˚C) | Mean temperature growing season (T˚C) | Mean temperature Mar-Sept (T˚C) |
|---|---|---|---|---|---|---|---|---|
| Current | Current | 1246 | 228 | 69 | 558 | 9.9 | 24.5 | 25.7 |
| RCP2.6 | GFDL | +52 | +2 | -2 | +30 | -0.4 | +1.5 | +1.3 |
| | IPSL | -43 | +3 | +3 | -14 | -0.2 | +1.8 | +1.5 |
| | HADGEM | -4 | +10 | +3 | +14 | -0.3 | +2.2 | +1.4 |
| | MIROC | +58 | +21 | +6 | +41 | -0.4 | +1.1 | +1.3 |
| | Model mean | +16 | +9 | +3 | +18 | -0.3 | +1.7 | +1.4 |
| RCP8.5 | GFDL | +59 | +27 | +3 | +40 | -0.7 | +2.1 | +2.5 |
| | IPSL | -117 | -3 | +2 | -29 | -0.1 | +2.8 | +2.6 |
| | HADGEM | -13 | +10 | +3 | +7 | -0.5 | +1.5 | +2.5 |
| | MIROC | +86 | +21 | +2 | +51 | -0.3 | +2.3 | +1.8 |
| | Model mean | +3.8 | +13.9 | +2.8 | +17.5 | -0.4 | +2.2 | +2.3 |

a single strong learner. This is achieved through sequentially building a series of smaller trees, where each tree tries to complement each other and correct for the residuals in the predictions made by all previous trees [53, 54]. The XGBoost algorithm develops the GBM approach further through using an ensemble of classification and regression trees (CARTs) to fit training data samples to targets [55, 56]. An independent binary tree decision rule structure is produced for each CART and contains a continuous score on each leaf node, and for a given input, the output is the sum of the corresponding leaves' scores [57, 58]. This is done in an additive way as the predictions are made from weak classifiers that constantly improve over the previous prediction error with higher weights at the next step to improve the scoring. The learning of the model in XGBoost is based on defining an objective function which describes the predictive accuracy of the model and regularization function describing the complexity [54, 55, 57]

The XGBoost approach was developed by Chen and Guestrin [54] and has been widely recognized as one of the best machine learning algorithms because it is fast, accurate and based on smaller models compared to similar family of models [55, 59]. It also has better results because it ameliorates the iterative optimization procedure inherent in traditional GBMs [54]. Specifically, for crop suitability modelling the XGBoost is able to provide a scaled four class model that is more representative than a binary suitable/unsuitable model in a computationally efficient way for large scale spatial analysis, and since it reduces over-fitting with feature subsampling, it is more appropriate for model extrapolation under climate change. In recent machine learning performance studies, XGBoost has outperformed other algorithms [56, 60–62]. Parameter tuning for automatically determining the number of rounds, maximum tree depth and sigma was done using the *caret* package [63] while the XGBoost was implemented with the *xgboost* package [64] using the *multi:softmax* objective in R (version 3.5). The input data was randomly split into 70% for model fitting (training and validation) and the remaining 30% for independent model testing.

## 2.5 Identifying the contribution of variables to crop suitability

We used the regularized gain to determine the contribution of each variable to suitability of each crop. The *gain* is the relative contribution of a variable to the model calculated by taking each variable's contribution for each tree in the model. A higher value of this metric when compared to another variable implies it is more important for generating a prediction. If the difference between the full model (with all variables) and a model without a specific variable is small, it is assumed that the relative importance of this variable is low and vice versa. The contribution of each variable is then standardized between 0 (lowest importance) and 1 (highest importance) [65]. This approach is widely used in selection of variables that are important for predicted variables in machine learning.

## 2.6 Model evaluation

We used the confusion matrix to assess the accuracy of the modelled suitability classes relative to reference data that was set aside for model evaluation. The overall accuracy (OA), kappa coefficient, multi-class AUC and class specific metrics (sensitivity, specificity, positive prediction value, negative prediction value, precision, recall, F1-score, prevalence, detection rate, detection prevalence and balanced accuracy) were used as calculated from the confusion matrix. OA is the percentage that indicates the probability that a grid cell is modelled correctly by the model relative to the known reference data. The OA is calculated by dividing the sum of the entries that form the major diagonal (i.e., the number of correct classes) by the total number of samples for each crop. The kappa coefficient ($k$) [66] measures the accuracy of the

model predictions by comparing it with the accuracy expected to occur by chance with $k$ values ranging from -1 (poor) to 1 (good) [22]. The multiclass area under receiver operating characteristic curve (AUC) was used to validate model fit by comparing and averaging all pairwise class AUC. Sensitivity for suitability class is the percentage of a category on the reference data that is correctly modelled as belonging to that category, and measures proportion of pixels omitted from a reference suitability class (omission error). Specificity expresses the proportion of a category on the reference data that is included erroneously in another suitability class (commission error) [67]. Other metrics used for class accuracy are described in full in literature [68, 69]. This analysis was done in R v.3.3 (R Core Team, 2013).

## 2.7 Assessment of climatic suitability for cultivation of multiple crops

In order to determine the climatic suitability for cultivation of the four key food crops for Ghana, we combined the suitability of the crops to understand which areas are suitable for which crops and to what degree. At first the maps were stacked to determine the number of crops that were suitable for each cell. To determine suitability for multiple crops, we summed the modelled crop suitability with each class ranked from 1 (limited) to 4 (optimal). This produced climatic suitability scale crops for the four crops on a scale from 4 (very low) to 16 (very high). After that, realizing that suitability of two-crops was most frequently observed, further analysis of determining which pairs of crops were suitable at each of the pixels. Pairs of crops were summed to produce a potential between 2 (very low) to 8 (very high). Changes in suitability proportion and distribution between the current and the projected climatic conditions were assessed by counting and comparing areas and proportions of cells between times and scenarios.

## 3. Results

### 3.1 Model performance evaluation

To reliably assess crop suitability, we first evaluated the fit of the model on an independent test data set. There were differences in the model fit between crops, but all crops showed a good fit. The best accuracy was for modelling sorghum (OA = 0.82, $k$ = 0.75 and AUC = 0.87) (see Table 3). Modelling evaluation metrics for each of the four suitability classes such as sensitivity, specificity, positive and negative prediction values, precision, recall, F1-score, prevalence, detection rate, detection prevalence and balanced accuracy are shown in S1 Table. These accuracy metrics indicated that the model was able to match observed classes for all the crops and thus could be used with confidence in assessing suitability of the four crops in Ghana under current and future climate.

### 3.2 Contribution of variables to crop suitability

The relative contribution of each variable to modelling crop suitability was determined by analyzing the importance of each variable to the model. The percent contributions of each variable

**Table 3. Overall accuracy, kappa and multi-class AUC values as indicators model performance for maize, sorghum, groundnut and cassava in determining crop suitability classes in Ghana.**

| Crop | Accuracy | Kappa | Multiclass-AUC |
|---|---|---|---|
| Maize | 0.75 | 0.66 | 0.81 |
| Sorghum | 0.82 | 0.75 | 0.87 |
| Groundnuts | 0.75 | 0.66 | 0.85 |
| Cassava | 0.71 | 0.59 | 0.84 |

to the explained variability of each crop are shown in Fig 2. The suitability of each crop and its geographical range are influenced by different biophysical parameters. The rainfall factors combined (sum of all three rainfall-related factors) have a larger influence on the potential suitability for the four crops in Ghana compared to the influence of temperature-based factors. These rainfall factors explain up to 60% of explained variability of cassava, up to 59% the variability of groundnuts, 66% for maize and 57% the variability of sorghum (Fig 2).

The total sum of rainfall received between March and September is more important in determining the suitability of maize (25%), cassava (23% and sorghum (19%). The mean temperature was not identified as important for any crop, but the diurnal temperature range explains about a quarter of the suitability for sorghum (26%), which is the highest value for any single variable. Variation of rainfall was important for all crops in almost equal measure (12–16%), with more importance for maize. Soil organic carbon was important (>10% contribution) for maize and cassava. The contribution of rainfall sum for sowing months, growing season mean temperature and mean temperature between March and September to the suitability of the four crops was mostly minimal (Fig 2).

## 3.3 Projected climate change impacts on agronomic variables

Projected changes in agronomically important variables are shown in Table 3. The results indicate an increase in rainfall in Ghana for the cropping period, the sowing month and the growing season. However, these increases in rainfall will be accompanied by increases in the CV of rainfall of 3% for both scenarios at national level. Less rainfall increases are projected for RCP8.5 compared to RCP2.6 except for rainfall amounts in the sowing months. There is GCM agreement in direction of change for rainfall CV for RCP8.5 and sum of rainfall for sowing months for RCP2.6 and fir all temperature variables, which all indicate warming (Table 3). As expected, higher warming is projected under RCP8.5 compared to RCP2.6 with a decreasing temperature range.

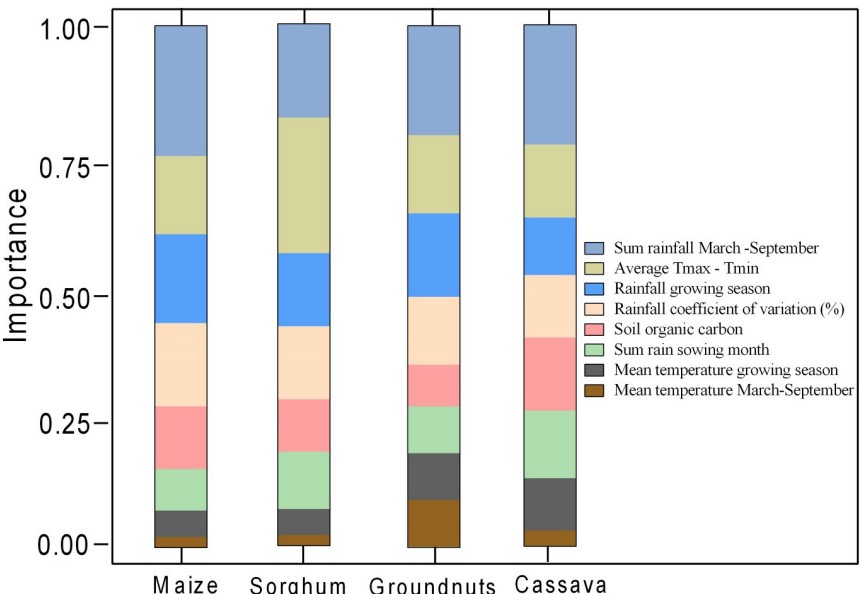

**Fig 2. Variable importance of each of the parameters used in determining the suitability for maize, sorghum, groundnut, and cassava in Ghana.**

## 3.4 Suitability and suitability changes of individual crops

Under current climatic conditions, the suitability of maize is very variable across the country with no specific regional distribution. Optimal suitable areas for maize cover 22% (51323 km$^2$) of the country (Figs 3A and 4A). Under projected climatic conditions the areas that have optimal suitability for maize production will decrease by 12% (6084 km$^2$) and by 14% (7171 km$^2$) under RCP2.6 and RCP8.5 respectively as suitability transition from being optimal to moderately suitable and marginal. These are the largest changes from the optimal suitable areas of the crops modelled in this study. Areas that have marginal suitability are projected to increase by 8% (6885 km$^2$) under RCP2.6 and by 7% (5703 km$^2$) under RCP8.5 scenario, with limited areas decreasing by 11% or 5800 km$^2$ (RCP2.6) and by 8% or 4508 km$^2$ (RCP8.5) (Figs 3B and 3C and 4A and Table 4).

Sorghum was modelled as having largest area for which it is optimally suitable (28% or 66731 km$^2$), which is the highest of the four crops for this category (Figs 3D and 4B). Under climate change, the optimal suitability areas for sorghum are projected to decrease by 10% (6445 km$^2$) and 13% (8716 km$^2$) decrease under RCP2.6 and RCP8.5 respectively (Figs 3E and 3F and 4B). Some parts of northern Ghana that have limited suitability for sorghum will become suitable under both RCP2.6 and RCP8.5 with an evident northwards shift in sorghum suitability under climate change. The areas that are unsuitable for sorghum are projected to increase by 12% or 7263 km$^2$ (RCP2.6) and 13% or 7601 km$^2$ (RCP8.5) by the 2050s (Fig 3E and 3F and Table 4).

Cassava was modelled as mostly suitable in the southern forested bimodal rainfall areas of Ghana (Fig 3G). Under RCP2.6, the results show that by the 2050s, optimal suitable areas for cassava will decrease by 7% (2798 km$^2$) while under RCP8.5, they will decrease by 9% (3731 km$^2$) (Fig 4C). Concurrently, the areas that have limited suitability for cassava will also slightly decrease by 4% (3039 km$^2$) under RCP2.6 and by 3% (2338 km$^2$) under RCP8.5 from the current 35%. The results showed that 48% (115868 km$^2$) of Ghana can produce groundnuts (optimal and moderate suitability) under current climatic conditions (Fig 3J). Optimal suitable areas (17% or 39861 km$^2$) are mostly located in the northern and central zones of Ghana (Fig 3K and 3L). Under projected climate change, the results show that the areas that have optimal suitability for groundnuts will decrease by 3% (1498 km$^2$) under RCP8.5 but will remain stable at 17% (+169 km$^2$) under RCP8.5, a trajectory different from other crops modelled in this study (Table 4 and Fig 4D).

## 3.5 Climatic suitability of multiple crops for current and future climatic conditions

The suitability of each pixel for multiple crops was evaluated by determining the number of crops in each suitability class suitable for that pixel. Under all climatic conditions, none of the areas has optimal suitability for all the four crops in Ghana (Fig 5A). Under current climatic conditions, 2% (3716 km$^2$) of the country has moderate suitability for at least three of the four crops. This area is projected to remain unchanged under both RCP2.6 and RCP8.5 (Table 5 and Fig 4B and 4C and S2 Table).

Similar significant decreases in suitability are also projected for areas that have moderate suitability for at least two of the four crops as these will decrease to 5% (12837 km$^2$) under RCP2.6 and 6% (13512 km$^2$) under RCP8.5. Much of these high suitable areas will become suitable for fewer crops under climate change as areas that are moderately suitable for all the four crops will increase to 1% under both RCP2.6 (3040 km$^2$) and RCP8.5 (3378 km$^2$) while those moderately suitable for two crops will increase from 19% (44591 km$^2$) to 24% or 57427 km$^2$ (RCP2.6) and 25% or 60130 km$^2$ (RCP8.5). Under both scenarios, the areas that are

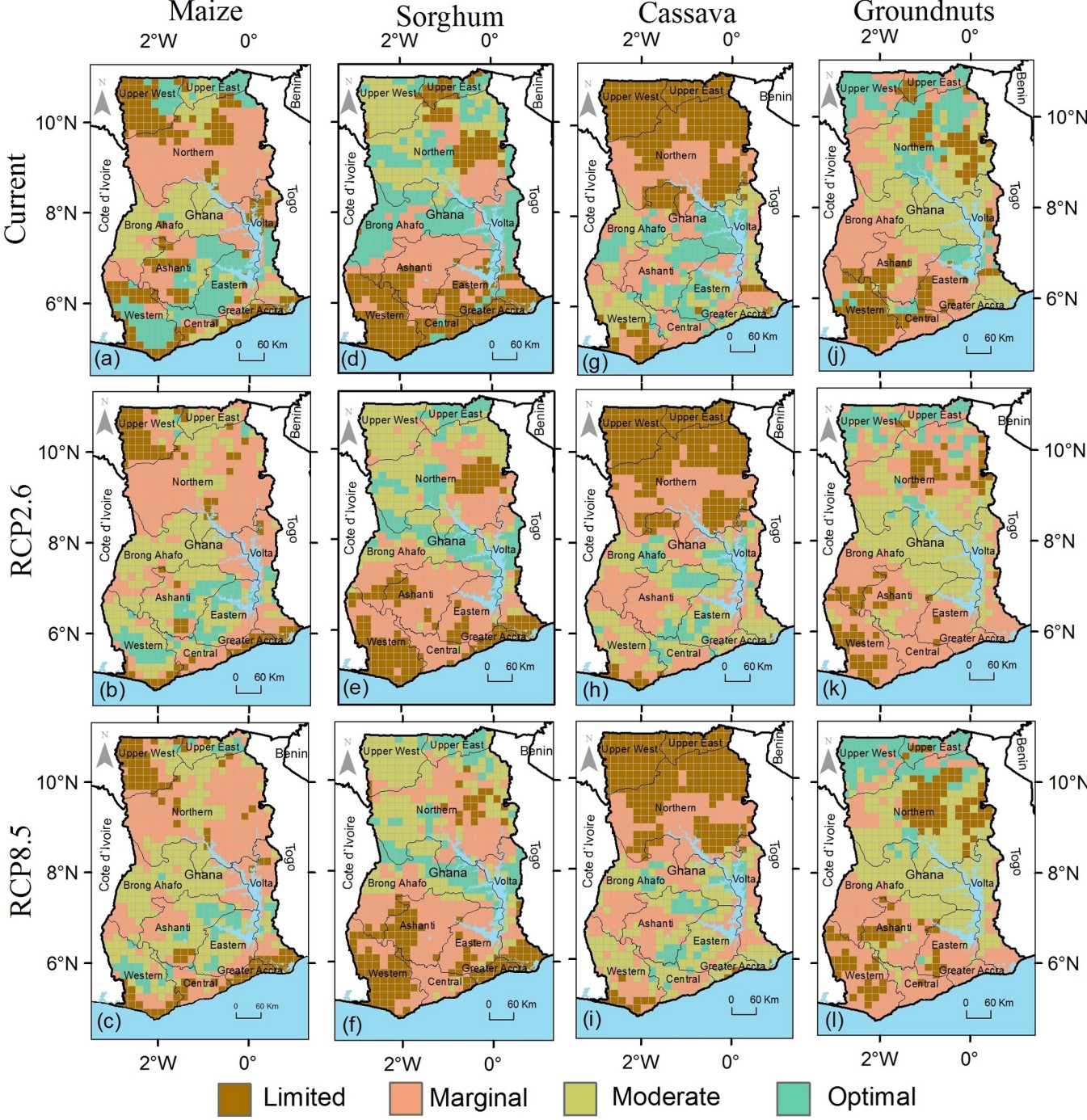

**Fig 3.** Suitability maps for (a) maize under current conditions (b) maize by 2050 under the RCP2.6, (c) maize under RCP8.5, (d) sorghum under current conditions (e) sorghum in 2050 under RCP2.6 in Ghana under (f) sorghum under RCP8.5, (g) cassava under current conditions (h) cassava in 2050 under RCP2.6 in Ghana under (i) cassava under RCP8.5, (j) groundnuts under current conditions (k) groundnuts in 2050 under RCP2.6 in Ghana under (l) groundnuts under RCP8.5.

marginally suitable for the four crops will increase from 1% (1689 km$^2$) under current climate to 3% (7770 km$^2$ RCP2.6) or 1% (3040 km$^2$ for RCP8.5) of the country. Similarly, the areas that are marginal for two of the four crops will increase from 9% (21958 km$^2$) under current

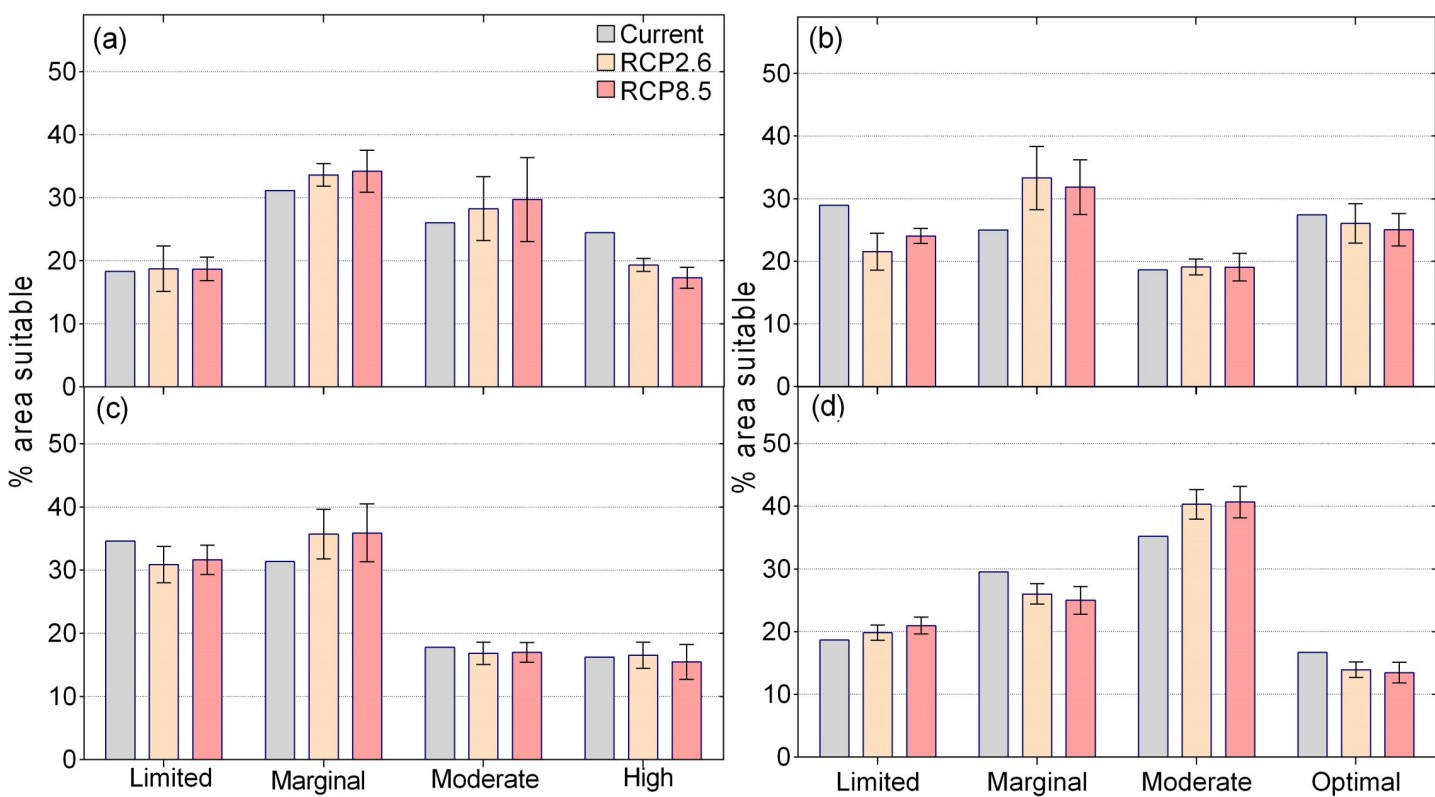

**Fig 4.** Assessment of changes in suitability according to the applied GCMs and RCP for (a) maize (b) sorghum, (c) cassava and (d) groundnuts in Ghana.

climatic conditions to 17% under RCP2.6 (39861 km$^2$) and RCP8.5 (40875 km$^2$), with a concurrent reduction in areas that are moderately suitable for just one crop as these become less (Fig 4B and 4C and Table 5).

We assessed the suitability of dual crops under current and projected climatic conditions through pairwise combinations of suitability maps (Fig 6). Areas with a highest suitability of dual crops are 5.5% (13175 km$^2$) for maize and groundnuts, 5.4% (12837 km2) for cassava and sorghum and 5.2% (12499 km$^2$) for maize and cassava (Figs 6 and 7 and S2 Table); all other combinations are below 5%. Except for cassava and groundnut, all suitability combinations of crops are projected to decrease for the areas where both crops currently have the high suitability class. Concurrently, the areas where a more crops will be moderate and marginal or marginal for both crops will increase for both RCP2.6 and RCP8.5 (Fig 7). Although the results show on dual crop suitability vary across the country, aggregated results show the most common suitability under climate change will be sorghum and groundnuts for the northern parts and cassava and groundnuts for the southern parts of Ghana. The least potential dual

**Table 4. Mean percentage changes in area suitable for each suitability class under climate change for maize, sorghum, cassava and groundnuts.**

| Crop | Maize | | Sorghum | | Cassava | | Groundnut | |
|---|---|---|---|---|---|---|---|---|
| Suitability | RCP2.6 | RCP8.5 | RCP2.6 | RCP8.5 | RCP2.6 | RCP8.5 | RCP2.6 | RCP8.5 |
| Limited | -10.6 | -8.2 | -11.8 | -9.4 | -3.7 | -2.8 | -7.6 | -7.1 |
| Marginal | 8.1 | 6.6 | 12.1 | 12.7 | 5.2 | 4.2 | 11 | 4.5 |
| Moderate | 14.4 | 15.5 | 9.3 | 9.7 | 5.6 | 8.2 | 0.4 | 2.1 |
| Optimal | -11.9 | -14.0 | -9.7 | -13.1 | -7.2 | -9.6 | -3.8 | 0.4 |

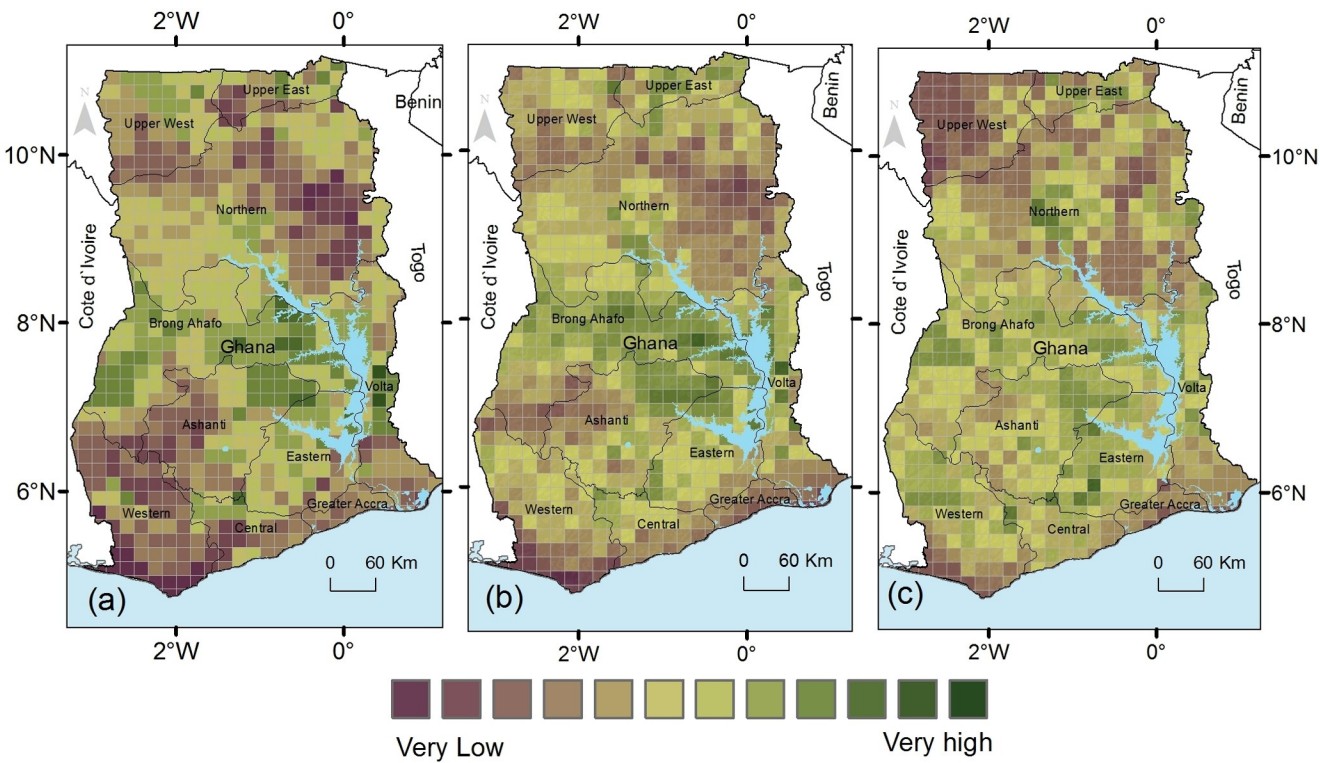

**Fig 5.** Modelled suitability maps of four crops under (a) current (b) RCP2.6 and (c) RCP8.5 climatic conditions for Ghana.

suitability will be for cassava and sorghum, and groundnut and sorghum. There are also differences in dual crop suitability between the north and south of Ghana (S3 Table).

## 4. Discussion

The impacts of climate change on food security are multi-faceted. Therefore, in this study we quantified the impacts of climate change on the multiple crop potential by assessing the suitability of key food crops in the case of Ghana. A model for estimating crop suitability for maize, sorghum, cassava and groundnuts under current climate was constructed, which reliably reproduced observed suitability patterns. Therefore, we deem the model as sufficiently robust to predict the suitability of the four crops under future climate conditions. We identified, quantified and mapped individual and multiple crop suitability for assessing impact areas projected climatic conditions.

### 4.1 Contribution of variables to crop suitability in Ghana

Important biophysical predictors of the suitability of each crop were identified and these correspond to the reported crop requirements, growing conditions and spatial distribution of the four crops in Ghana [70–72]. The finding that precipitation-based factors are most important for the suitability of maize, sorghum and cassava is in line with other studies as rainfall remains the most important determinant of agricultural production in many African countries. For example, drought stress and related plant water availability constraints have been singled out as the most limiting factors for these crops in West Africa and elsewhere[8, 73–76], particularly as these crops are almost entirely produced under rain-fed conditions.

**Table 5. Area and percentage of different levels of suitability of multiple crops in Ghana under current and future climates.**

| Suitability | Scenario | Measure | One crop | Two crops | Three crops | Four crops |
|---|---|---|---|---|---|---|
| Very high | Current | Area(km$^2$) | 98978 | 41888 | 3716 | 0 |
| | | % | 41 | 18 | 2 | 0 |
| | R26 | Area(km$^2$) | 69588 | 12837 | 338 | 0 |
| | | % | 29 | 5 | 0 | 0 |
| | R85 | Area(km$^2$) | 79385 | 13512 | 338 | 0 |
| | | % | 33 | 6 | 0 | 0 |
| High | Current | Area(km$^2$) | 105058 | 44591 | 5405 | 0 |
| | | % | 44 | 19 | 2 | 0 |
| | R26 | Area(km$^2$) | 107761 | 57427 | 20268 | 3040 |
| | | % | 45 | 24 | 8 | 1 |
| | R85 | Area(km$^2$) | 104045 | 60130 | 18917 | 3378 |
| | | % | 44 | 25 | 8 | 1 |
| Low | Current | Area(km$^2$) | 89519 | 65197 | 21958 | 1689 |
| | | % | 37 | 27 | 9 | 1 |
| | R26 | Area(km$^2$) | 87492 | 77696 | 39861 | 7770 |
| | | % | 37 | 32 | 17 | 3 |
| | R85 | Area(km$^2$) | 62832 | 62494 | 40875 | 3040 |
| | | % | 26 | 26 | 17 | 1 |
| Very low | Current | Area(km$^2$) | 74318 | 59792 | 13175 | 1351 |
| | | % | 31 | 25 | 6 | 1 |
| | R26 | Area(km$^2$) | 61143 | 38510 | 3716 | 3716 |
| | | % | 26 | 16 | 2 | 2 |
| | R85 | Area(km$^2$) | 77696 | 29389 | 14188 | 0 |
| | | % | 32 | 12 | 6 | 0 |

Although rainfall amounts are generally high in Ghana (over 800 mm in most years), precipitation remains an important factor in determining crop potential. This is because of the climate gradient in Ghana from the south to the north, and its intra-seasonal variation influencing the suitability of the different crops. For comparison, in China fit was observed that temperature-related variables as more important factors for maize suitability than precipitation [77], highlighting the local relevance of our results that cannot simply be extrapolated. We found that sorghum suitability is also influenced by the diurnal temperature range, which concurs with current understanding that sorghum is a more adverse weather tolerant crop. This tolerance is enabled by heterotic mechanisms that allow for greater biomass and yield production at a shorter period, 'stay green' mechanisms, and lodging and desiccation tolerance compared to other crops [78, 79].

## 4.2 Individual and multiple crop suitability under climate change

Of the four crops modelled in this study, groundnut suitability is the most resilient under climate change, showing the smallest loss in suitable growing areas. This could be explained by groundnuts being legumes with a short growth period and whose harvested parts grow below ground and thus are partly protected by the soil from direct effects of warming. Groundnut viability is closely related to rainfall patterns [80, 81], particularly the amount of rainfall in the growing season due to the less extensive root system, and thus projected increases in rainfall can directly increase suitability for groundnut production. This is especially so as the assessment of changes in agronomic variables show increased rainfall variability than changes in

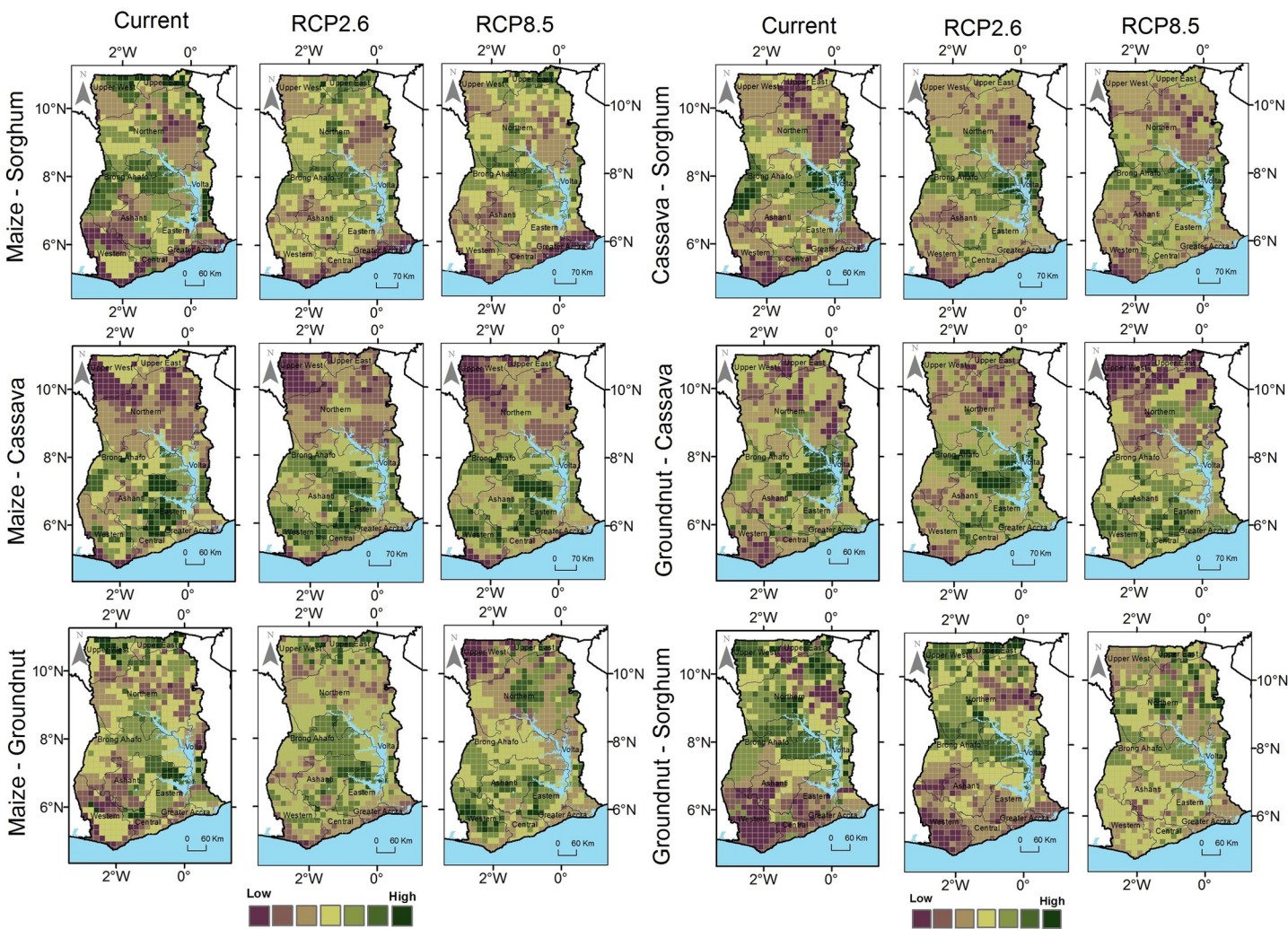

**Fig 6. Maps showing the suitability of dual crop suitability for pairwise crop combinations across Ghana under current, RCP2.6 and RCP8.5 climatic scenarios.**

total values. These results concur with findings by earlier studies on the potential impacts of climate change in Ghana which reported groundnuts as less impacted compared to other food crops [5]. Our findings show that sorghum remains a high potential crop in the northern parts of Ghana under a changed climate, as limited areas in these areas decrease. This result underlines the importance of sorghum, which is already a major food crop in the northern parts of Ghana.

The greatest climate change risk was identified for maize which is the crop with the most planted area and the highest net consumption in the country [82]. Apart from the reliance of maize production on rainfall in Ghana, the fact that maize is more sensitive to weather variables than other crops also explain this loss. Maize responds to both warming and increased rainfall variability as water deficit can cause reduced growth by allocating more carbon to the root system, reducing leaf expansion and photosynthesis. Higher temperature, meanwhile, can cause loss of pollen viability, damage to tissue enzymes and accelerated senescence [83]. These severe impacts of climate change on maize production in Africa have already been reported elsewhere [46, 84, 85].

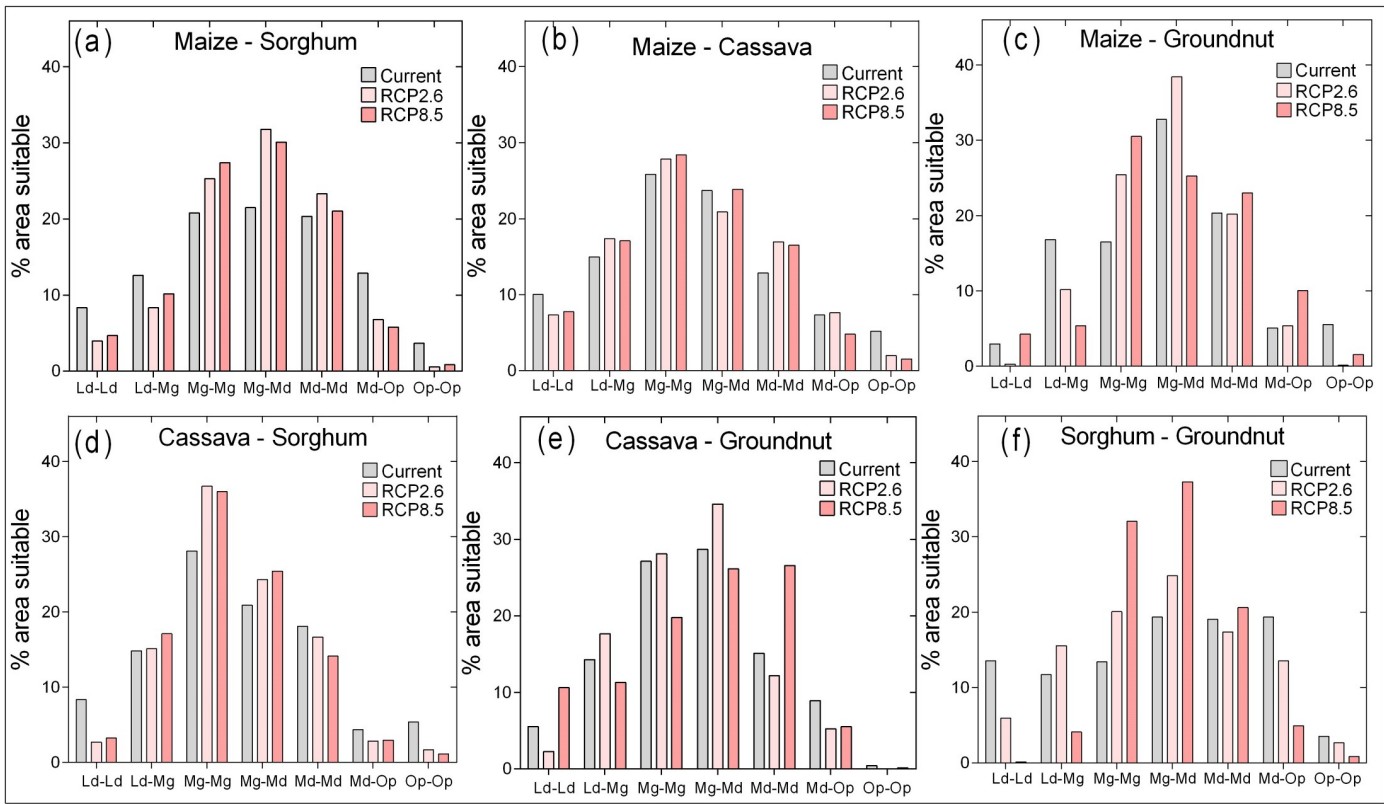

**Fig 7.** Area fractions suitable for (a) maize and sorghum (b) Maize and cassava (c) Maize and Groundnut (d) Cassava and Sorghum, (e) cassava and groundnut and (f) Sorghum and groundnut. The lines are Gaussian distribution fit for each climatic scenario. Ld-Ld is for limited suitability for both crops, Ld-Mg is area with limited for one of the 2 crops and marginal for the other, Mg-Mg is area where both crops are marginal, Mg-Md is area that is marginal for one crop while moderately suitable for the other crop, Md-Md is where both crops are moderately suitable, Md-Op is where one crop is moderately suitable and the other has optimal suitability and Op-Op is where both crops are optimally suitable.

In addition to these crop-specific climate responses, the predominant outcome of the suitability modeling is that the impacts of climate change are site and crop-specific. The impacts are determined by both the biophysical factors that influence crop viability and the specific genetic characteristics of the crops. Shifts in crop suitability have been identified as a key influence of climate change, spurring a need for adequate adaptation measures in the identified areas or planning for food transfer systems that distribute food between the areas that will become suitable and those that will become marginal for a particular crop [19, 86].

This study indicates that there will be a reduction in the suitability of multiple crops in Ghana under climate change. Areas which are currently optimally and moderately suitable for production of the four crops will decrease while areas marginally and moderately suitable for two or more crops will increase. Although multiple crop suitability does not indicate potential for growing the crops together, the reduced potential for multiple crops under climate change reduces the choices farmers have in terms of crop production, which increases risks. The more the crops are suitable for a farmer in a pixel, the mode the farmer can make production choices for crops for both food security and trading of surplus, which are both being curtailed by climate change. Furthermore, the finding that two crop combinations involving sorghum and groundnuts for the north and maize and groundnuts for the south have the most potential under climate change is important for adaptation planning and investment in these crops.

There are a number of adaptation measures that can be applied to increase multiple crop suitability to avert the modelled reduced potential. Altieri and Nicholls [87] posit that smart agricultural systems such as raised beds and semi-permanent water collecting basins that act as field-scale micro-catchments can sustain production of different crops under climate change as they can work for most crops. These and other conservation agriculture techniques could improve potential for multiple crops. This is especially important as rain fed agricultural systems are projected to remain dominant in African agricultural systems [88]. While these have potential, they should not add to complexity of already complex agricultural systems in Ghana and other African countries [89] through, for example, increasing labor burden [90, 91]. At national scale, policies encouraging the production of multiple crops such as development of market incentives for many crops can also help promote resilience. Shifting extension advice from individual crops to multiple crops could be an important first entry point.

### 4.3 Considerations in the interpretation of the results of crop suitability modelling

There are some limitations and potential sources of uncertainty that should be considered in the interpretation of our results. The suitability models are driven by climate and soil data and current crop production data, which have inherent uncertainties. Future projections of crop production suitability are produced by combining suitability models with projections based on GCMs that describe potential future conditions. These different GCMs rely on different parameters and incorporate different functions to cover the dynamics of atmospheric circulation, ocean effects, or feedbacks between the land surface and the atmosphere. Therefore, they are prone to disagreements or errors that will be propagated in the modelling. Our modelling omitted direct physiological interaction effects in multiple crops such as nitrogen fixation, water retention, pollination, completion or competition for nutrients that cannot be captured by this type of modelling. The area suitability calculations also include other land that may not be available for agricultural production because of the resolution. These other land areas are, for instance, urban areas, protected areas and riparian zones which cannot be removed at the spatial resolution of the datasets used. Thus, interpretation should be on the relative change rather than the absolute change in area suitable for each crop.

## 5. Conclusion

In this study we provide a quantitative starting point to gauge future suitability of multiple crops as a strategy to build climate resilient agricultural systems that are not available elsewhere. We conclude that impacts of climate change on different crops, regions and climatic scenarios are uneven, and highlight the crops and areas that are likely to be impacted the most. From such information, the types, scale and urgency of investing into adaptation strategies in the light of the NDC and NAP implementation process can be guided accordingly. Thus, our approach identified local impacts across the entire country that could be more useful for adaptation planning in ecologically, culturally and socio-economically heterogeneous farming systems such as in Ghana. In addition, and maybe more importantly, our integrated approach that assessed crop suitability of multiple crops implicitly captured opportunities and losses of many co-benefits that can be gained from multiple crops which are common in tropical countries but are not captured or indicated in individual crop assessments. The results from this study provide a scientific basis on which a national-level risk assessment on the impacts of climate change on multiple crops can be implemented. Since the information in this study is spatially explicit, areas requiring prioritization in adaptation action can be identified as those to experience the largest changes in area suitability and number of suitable crops. There is a

chance that the impacts of climate change could be reduced through systematic planning for climate-adapted development. This is, to our best knowledge, the first study to assess the impacts of climate change on multiple crops quantitatively at this resolution for a whole country.

## Supporting information

**S1 Fig. Map showing the climatic division for north and south in Ghana.**
(TIF)

**S1 Table. Accuracy metrics for modelling maize, sorghum, groundnut and cassava suitability in Ghana under current climate conditions.**
(DOCX)

**S2 Table. Percentage of dual crop suitability under each climatic condition in Ghana derived from combining the suitability levels of each of the two crops.**
(DOCX)

**S3 Table. Assessment of dual crop suitability for the north and south of Ghana using pairwise comparisons.**
(DOCX)

## Acknowledgments

We are grateful to Lisa Murken for project support and Stephanie Gleixner for assisting with climate data.

## Author Contributions

**Conceptualization:** Abel Chemura, Christoph Gornott.

**Data curation:** Abel Chemura.

**Formal analysis:** Abel Chemura, Bernhard Schauberger, Christoph Gornott.

**Funding acquisition:** Christoph Gornott.

**Investigation:** Abel Chemura, Bernhard Schauberger, Christoph Gornott.

**Methodology:** Abel Chemura, Bernhard Schauberger, Christoph Gornott.

**Project administration:** Abel Chemura, Christoph Gornott.

**Software:** Abel Chemura.

**Supervision:** Christoph Gornott.

**Writing – original draft:** Abel Chemura, Bernhard Schauberger.

**Writing – review & editing:** Abel Chemura, Bernhard Schauberger, Christoph Gornott.

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
