## [Decision Letter · Decision Letter 0]

30 Apr 2020

PONE-D-20-04571

Impacts of climate change on agro-climatic suitability of major food crops and crop diversification potential in Ghana

PLOS ONE

Dear Dr. Chemura,

Thank you for submitting your manuscript to PLOS ONE. After careful consideration, we feel that it has merit but does not fully meet PLOS ONE’s publication criteria as it currently stands. Therefore, we invite you to submit a revised version of the manuscript that addresses the points raised during the review process.

ACADEMIC EDITOR: 

I have received the review reports of your submitted manuscript. The reviewers have recommended major revisions.The methodlogy section of the manuscript has been criticized by the reviewers. There are major flaws in it; for example, different predictors have been used for different scenarios and the scenarios have been compared with each other. How could results from different predictors compared across scenarios?The other major reservation is the use of word "crop diversification". The methodlogy has predicted the suitability of individual crops, no co-occurrences have been modeled in the current manuscript. Thus, it is simple suitability for different crops, not crop diversification.Numerous references cited in the MM section have used Maxent, while the used methods are different than Maxent. Therefore, cite proper references.Please revise your manuscript carefully keeping in v,ew the comments of both reviewers.

We would appreciate receiving your revised manuscript by Jun 14 2020 11:59PM. To enhance the reproducibility of your results, we recommend that if applicable you deposit your laboratory protocols in protocols.io, where a protocol can be assigned its own identifier (DOI) such that it can be cited independently in the future. For instructions see: http://journals.plos.org/plosone/s/submission-guidelines#loc-laboratory-protocols

We look forward to receiving your revised manuscript.

Kind regards,

Shahid Farooq, Ph.D.

Academic Editor

PLOS ONE

2. We note that Figures 3, 5 and 6 in your submission contain map images which may be copyrighted. All PLOS content is published under the Creative Commons Attribution License (CC BY 4.0), which means that the manuscript, images, and Supporting Information files will be freely available online, and any third party is permitted to access, download, copy, distribute, and use these materials in any way, even commercially, with proper attribution. For these reasons, we cannot publish previously copyrighted maps or satellite images created using proprietary data, such as Google software (Google Maps, Street View, and Earth). For more information, see our copyright guidelines: http://journals.plos.org/plosone/s/licenses-and-copyright.

1.    You may seek permission from the original copyright holder of Figures 3, 5 and 6 to publish the content specifically under the CC BY 4.0 license. 

4. Please upload a copy of Supporting Information Table 4 which you refer to in your text on page 9.

Reviewers' comments:

Reviewer's Responses to Questions

**Comments to the Author**

1. Is the manuscript technically sound, and do the data support the conclusions?

Reviewer #1: Partly

Reviewer #2: Partly

2. Has the statistical analysis been performed appropriately and rigorously? 

Reviewer #1: No

Reviewer #2: No

3. Have the authors made all data underlying the findings in their manuscript fully available?

Reviewer #1: No

Reviewer #2: Yes

4. Is the manuscript presented in an intelligible fashion and written in standard English?

Reviewer #1: Yes

Reviewer #2: Yes

5. Review Comments to the Author

Reviewer #1: The manuscript addresses the influence of climate and climate change with different types of cultures that have food importance for Ganna. After many revisions in the sample design and mainly in the methodology, the manuscript deserves to be published. One of the main concerns regarding the research content is the use of this methodology to assess whether the crops can be grown together. The text, the methodology and the results point to places (pixels) where the crops can be grown together, but the variables analyzed were only the climatic ones. Variables related to aspects of co-occurrence were not included, which would explain whether the species can be cultivated together. An alternative is to include variables that may allow analyzing the co-occurrence of species in the same pixel, or change the approach from "combined suitability", "crop diversification" and "combined suitability" to "climatic suitability for cultivation". Major and minor revisions follow in more detail in each session.

Major revisions

Introduction

line 78 - 80 I suggest not using the term “combined suitability”. The methodology and analyzes do not make it possible to assess the potential of combined cultivation

linha 85 - 87 As análises permitem apenas “assess the impacts of projected climate change on four important food crops in Ghana by mid-century”. “Their ability to be produced together” não é possível de avaliar com essa metoologia e dados.

line 87 - 90 (i) ok. (ii) The analyzes allow only “identify climate change impacts on crop climatic suitability for individual crop”. (iii) Pixel overlay alone does not allow “determine crop diversification opportunities” (see MORUETA ‐ HOLME, N .; BLONDER, B .; SANDEL, B .; MCGILL, BJ; PEET, RK; OTT, JE; SVENNING, JC A network approach for inferring species associations from co-occurrence data (Ecography, v. 39, n. 12, p. 1139-1150, 2016).

Methodology

2.2 Show here that the objective is to find the pixels within the study area for each different class (optimal suitability, moderate suitability...)

- What parameters you used to determine why a pixel belongs to a particular class. How did you choose these criteria? Did you use, for example, the ideal amount of precipitation for the cultivation of each crop?

2.3 You must use the same variables both in the scenario between 2016 and 2016 (table 01) and in the scenario 2041 to 2050 (table 2). If you use different variables for each scenario, it is not possible to compare the models between the scenarios. If you used the same variables then make that clear.

2.4 References {28, 55-57,60,62} use models of maximum entropy (maxent). These models are different from yours. Add more references that use the same model you used.

- Explain why the XGBoost approach is better than the approaches cited in the text {28, 55-62}. Or use a reference that has already compared the best approach.

- Provide a brief explanation of the XGBoost method. How does it correlate the input data?

2.5 What is a full model?

- This section can be called “identifying the contribution of variables”

2.6 For a post analysis, use the areas where these crops are already grown to assess whether the “optimal suitability” class was effective in predicting areas where each crop already exists, for example. Use a polygonal cultivation shapefile for each crop for this step. Use this analysis to show the percentage of pixels in the “optimal suitability” class that are found within the polygon where the crop already exists, for example.

For this it is possible to use the same criteria used in the evaluation of the models.

2.6 This session should be 2.7.

- Research to make predictions for the fitness area for more than one species use the stacking of models (See CALABRESE, J.M.; CERTAIN, G.; KRAAN, C.; DORMANN, C.F. Stacking species distribution models and adjusting bias by linking them to macroecological models. Global Ecology and Biogeography, v. 23, n. 1, p. 99-112, 2014) or pixels as is the case with your methodology. This makes stacked models tend to predict many species per location (See GUISAN, Antoine; RAHBEK, Carsten. SESAM - uma nova estrutura que integra modelos macroecológicos e de distribuição de espécies para prever padrões espaço-temporais de assembléias de espécies. Journal of Biogeography , v. 38, n. 8, p. 1433-1444, 2011).

3 Results

Include a description of all figures and tables

- Include the north on the maps

- Make available in the supplementary material all the results of the model output files.

3.3 Calculate in km² or he the areas both current and future scenarios. This gives you a more realistic view of the data.

- Where's table 4? In the text it speaks in table 3 and table 5.

3.4 The input data may not be sufficient to say whether the crops can be grown in a paired way.

- If the current and future model were built with different variables, they cannot be compared.

- The results indicate that a pixel is adaptable for one or more of a species, however it does not indicate that the species can be cultivated in a combined way.

- Use another term or approach. With these results it is not possible to state that the species can be cultivated together or in a combined way. The results show which pixels have climatic suitability for one or more species.

4.2 Make it clear that the results do not assess whether a species is resilient to climate change, the results show the analysis of pixels suitable for the species.

Minor revisions

Standardize as references in the text. Put () or {}

Reviewer #2: I have evaluated the manuscript “PONE-D-20-04571, "Impacts of climate change on agro-climatic suitability of major food crops and crop diversification potential in Ghana"

It is a well planned study; however, there are many flaws in methodology. The authors have used only climatic variables, while soil and disturbance variables have been ignored. Crop diversification is the collective effect of climatic and soil factor. More specifically genetic factors as well. Therefore, I suggest to revise the methodology.

The other major flaw is use of different variables for different scenarios. In such cases models could not be compared across scenarios.

The methods used has assessed the individual suitability of crops not collective suitability

Calculate in km² or he the areas both current and future scenarios

Use another approach. With current results it is not possible to state that the species can be cultivated

Follow journal guidelines

6. PLOS authors have the option to publish the peer review history of their article (what does this mean?). If published, this will include your full peer review and any attached files.

Reviewer #1: No

Reviewer #2: No

---

## [Author Response · Author response to Decision Letter 0]

27 May 2020

Editor comments 

• Noted. Manuscript title and body formatted following the samples. 

2. We note that Figures 3, 5 and 6 in your submission contain map images which may be copyrighted. All PLOS content is published under the Creative Commons Attribution License (CC BY 4.0), which means that the manuscript, images, and Supporting Information files will be freely available online, and any third party is permitted to access, download, copy, distribute, and use these materials in any way, even commercially, with proper attribution. For these reasons, we cannot publish previously copyrighted maps or satellite images created using proprietary data, such as Google software (Google Maps, Street View, and Earth). For more information, see our copyright guidelines: http://journals.plos.org/plosone/s/licenses-and-copyright.

• All maps were produced by authors and not subject to copyrights. 

• Comment noted. All supporting information files now prefixed with “S” as required and listed at the end of the manuscript. 

4. Please upload a copy of Supporting Information Table 4 which you refer to in your text on page 9. 

• The table referred to on page 9 is now S2 Table Reference to supporting materials changed to new naming convention for supporting materials. 

5. We note that you have stated that you will provide repository information for your data at acceptance. Should your manuscript be accepted for publication, we will hold it until you provide the relevant accession numbers or DOIs necessary to access your data. If you wish to make changes to your Data Availability statement, please describe these changes in your cover letter and we will update your Data Availability statement to reflect the information you provide Data repository available at (https://zenodo.org/record/3669955/) and publicly available (DOI: 10.5281/zenodo.3669955). 

• Data availability statement updated to indicate that the data is now publicly available. 

Reviewer #1: 

The manuscript addresses the influence of climate and climate change with different types of cultures that have food importance for Ganna. After many revisions in the sample design and mainly in the methodology, the manuscript deserves to be published. One of the main concerns regarding the research content is the use of this methodology to assess whether the crops can be grown together. The text, the methodology and the results point to places (pixels) where the crops can be grown together, but the variables analyzed were only the climatic ones. Variables related to aspects of co-occurrence were not included, which would explain whether the species can be cultivated together. An alternative is to include variables that may allow analyzing the co-occurrence of species in the same pixel, or change the approach from "combined suitability", "crop diversification" and "combined suitability" to "climatic suitability for cultivation". 

• We thank the reviewer for the positive feedback on our submission. The point on toning down diversification to climatic suitability was noted corrected in the whole manuscript. We have changed many areas of the manuscript from diversification to climatic suitability for cultivation for the crops including changing the title of the submission to reflect this. We have adopted suggested climatic suitability for cultivation of multiple crops. 

Introduction

line 78 - 80 I suggest not using the term “combined suitability”. The methodology and analyzes do not make it possible to assess the potential of combined cultivation.

• This comment has been noted and implemented. Changed “combined suitability” and “diversification” to suitability for multiple crops.

linha 85 - 87 As análises permitem apenas “assess the impacts of projected climate change on four important food crops in Ghana by mid-century”. “Their ability to be produced together” não é possível de avaliar com essa metoologia e dados. 

• Agreed and done. Removed “Their ability to be produced together”

line 87 - 90 (i) ok. (ii) The analyzes allow only “identify climate change impacts on crop climatic suitability for individual crop”. (iii) Pixel overlay alone does not allow “determine crop diversification opportunities” (see MORUETA ‐ HOLME, N .; BLONDER, B .; SANDEL, B .; MCGILL, BJ; PEET, RK; OTT, JE; SVENNING, JC A network approach for inferring species associations from co-occurrence data (Ecography, v. 39, n. 12, p. 1139-1150, 2016). 

• We agree, as above, that combining the individual suitability results per crop does not yet allow for concluding on multi-cropping with its much more diverse interactions. We have therefore changed the whole manuscript to deliver results for the suitability of single crops for each pixel, which can be considered together and thus form a prerequisite for co-cultivation. The whole manuscript was changed accordingly to adequately judge results on combinations of individual crop suitability rather than diversification.

Methodology

2.2 Show here that the objective is to find the pixels within the study area for each different class (optimal suitability, moderate suitability...)

• Done

- What parameters you used to determine why a pixel belongs to a particular class. How did you choose these criteria? Did you use, for example, the ideal amount of precipitation for the cultivation of each crop?

• We used average production data from 2006 and 2016 to obtain 4 crop potentials classes and then fit an empirical multinomial model with climatic variables (total rainfall in the growing season, total rainfall received between March and September, sum of rainfall in the crop sowing month etc.). So this is a supervised model developed from reported yields and agronomic variables. We chose to use the percentiles of mean yields as primary classifying variable since the obtained yield is a good indicator of overall agronomic potential within a region. How much this potential is shaped by climate is the second step, estimated with the empirical model. We have included one further sentence in section 2.2 to emphasize that suitability assignments are done on pixel basis. We have added more information to make the model fitting descriptions more understandable to the reader from Line 125. 

2.3 You must use the same variables both in the scenario between 2016 and 2016 (table 01) and in the scenario 2041 to 2050 (table 2). If you use different variables for each scenario, it is not possible to compare the models between the scenarios. If you used the same variables then make that clear. 

• Exactly the same variables were used for the current and all scenarios for future projections. We have added in Line 139 that same variables were used for current and future climate projections to make this clearer to the reader. 

2.4 References {28, 55-57,60,62} use models of maximum entropy (maxent). These models are different from yours. Add more references that use the same model you used.

• Added references that used similar model as ours. 

- Explain why the XGBoost approach is better than the approaches cited in the text {28, 55-62}. Or use a reference that has already compared the best approach. Provide a brief explanation of the XGBoost method. How does it correlate the input data? 

• The major advantage of our approach is that instead of having only 2 suitability classes, we obtain a scaled four class model that is more informative and does not overestimate changes. More references for our approach added in Line 193-195. References that compared the best approaches and found the XGBoost as the best performing have also been added.

• 

-We have provided more information about why the XGBoost is a good approach for our model in Line 212-215 compared to the traditional [1,0] suitability approach.

• More information about how XGBoost works and relates target variables with input data has been provided in more detail in Line 198-209.

2.5 What is a full model?

• Full model is a model with all parameters. This information has been provided. 

- This section can be called “identifying the contribution of variables” The concept of the full model relates to how variable importance is calculated. The model is run with all variables (full model) and then in subsequent runs one parameter is removed to assess how much the fit changes and this change is used to show relative importance of each variable. 

• Done. We have improved the description on the identification of the important variables from Line 256-264.

• Section changed to Identifying the contribution of variables to crop suitability. 

2.6 For a post analysis, use the areas where these crops are already grown to assess whether the “optimal suitability” class was effective in predicting areas where each crop already exists, for example. Use a polygonal cultivation shapefile for each crop for this step. Use this analysis to show the percentage of pixels in the “optimal suitability” class that are found within the polygon where the crop already exists, for example.

For this it is possible to use the same criteria used in the evaluation of the models. 

• We thank the reviewer for this suggestion of additional validation. With an appropriate crop mask that does not only show existence but also suitability of a crop on high resolution, this would be a welcome evaluation. Yet, the cropland layer available just indicates croplands without showing if it is in “high”, “moderate”, “marginal” or “limited” class. This is because the yield data is from both high and limited areas. The pixel resolution of 25x25km is also difficult to narrow down to field polygons as no yield data is available at field level. Finally, we would like to emphasize that model calibration and validation are done with the same criteria. We have updated the description of the evaluation approach in Line 2.6 to make it more understandable. 

2.6 This session should be 2.7.

- Research to make predictions for the fitness area for more than one species use the stacking of models (See CALABRESE, J.M.; CERTAIN, G.; KRAAN, C.; DORMANN, C.F. Stacking species distribution models and adjusting bias by linking them to macroecological models. Global Ecology and Biogeography, v. 23, n. 1, p. 99-112, 2014) or pixels as is the case with your methodology. This makes stacked models tend to predict many species per location (See GUISAN, Antoine; RAHBEK, Carsten. SESAM - uma nova estrutura que integra modelos macroecológicos e de distribuição de espécies para prever padrões espaço-temporais de assembléias de espécies. Journal of Biogeography , v. 38, n. 8, p. 1433-1444, 2011).

• Noted. We have read with interest the concept of stacked SDM (S-SDM) from reference. As indicated by the reviewers, the input data does not tell if the same point has 1, 2 or 3 crops being produced, therefore we keep our analysis to the development of a multiple crop suitability index that does not show crop interactions. 

• -Corrected to Section 2.7.

• We have removed all references to crop diversification or pairing and focused on multiple crop suitability analysis. 

3 Results

Include a description of all figures and tables

• We have added section 3.3 in the results to describe the results of Table 3. We have also ensured also figures and tables are well described in text.

- Include the north on the maps

• North arrows added.

- Make available in the supplementary material all the results of the model output files. Table 3 was not described as it was originally put as a reference. 

• We have now ensured that all tables are described in the results. 

• We have made them available in the supplementary materials data on model performance, percentage and area suitable for north and south and percentage and area for dual crop suitability. 

3.3 Calculate in km² or he the areas both current and future scenarios. This gives you a more realistic view of the data. 

• Done.

- Where's table 4? In the text it speaks in table 3 and table 5. 

• Table 4 was converted to a figure and the numbering was not updated. Now corrected Area calculations added to the results and values added in the supplementary materials.

• All table numbering corrected and updated. 

3.4 The input data may not be sufficient to say whether the crops can be grown in a paired way. 

• Noted. Now we refer to this simply as dual crop suitability to indicate areas where two crops are suitable BUT without indicating if they can be grown together. 

If the current and future model were built with different variables, they cannot be compared. 

• Same variables were used for current and future to make the predictions comparable. We don’t understand how reviewers got this impression that different variables were used. Methods section updated to indicate that the same variables were used for current and future climatic conditions in Line 172.

The results indicate that a pixel is adaptable for one or more of a species, however it does not indicate that the species can be cultivated in a combined way. Use another term or approach. 

• We have now adopted the term multiple crop suitability with the understanding of the limits to saying the crops can be produced in a combined way. Changed from dual cropping to dual suitability and multiple suitability throughout the manuscript.

With these results it is not possible to state that the species can be cultivated together or in a combined way. The results show which pixels have climatic suitability for one or more species. 

• Corrected to indicate that pixel indicate the number of species that it is adaptable to, and not combined cultivation. 

• The term multiple crop suitability now applied throughout the manuscript to indicate that the pixels have suitability for one or multiple crops instead of combined or diversification. 

4.2 Make it clear that the results do not assess whether a species is resilient to climate change, the results show the analysis of pixels suitable for the species. 

• We thank the reviewer for this important hint. This passage is now obsolete with the change from diversity to multiple suitability. Resilience is now only used in conjunction with agricultural systems, not individual crops. Corrected to limit the descriptions to suitability. 

Minor revisions: Standardize as references in the text. Put () or {} 

• Done, this was caused by the reference managing app, now corrected. Standardized all references to match the publication requirement. 

Reviewer #2: 

It is a well planned study; however, there are many flaws in methodology. The authors have used only climatic variables, while soil and disturbance variables have been ignored. Crop diversification is the collective effect of climatic and soil factor. More specifically genetic factors as well. Therefore, I suggest to revise the methodology. 

• We thank the reviewer for the, in general, positive reception of our study. We also appreciate the clear suggestions how to improve our analysis and have followed these. More specifically, we have added more clearly that conclusions on diversification as intertwined genetic and species interaction effects should not be drawn from our model. 

• We also acknowledge that considering climatic variables alone does not yield the full picture of suitability in current or future times and have added a passage to the discussion. Yet our results indicate that climate exerts a major influence on suitability, and soil or other factors are more difficult to project for the future than climate – that is why we restricted our study to climatic impacts on suitability only. We have limited the study to climatic suitability. But we control also for soil organic carbon as one important soil variable.

The other major flaw is use of different variables for different scenarios. In such cases models could not be compared across scenarios.

• We certainly agree that the same set of variables needs to be used to judge on suitability in current and future times, and we have done so, using the exact same variables to ensure that the results are comparable. The same soil factor was used for current and future climate. We have corrected the manuscript to make it clear that the same variables were used for the different scenarios to ensure the results are comparable. 

The methods used has assessed the individual suitability of crops not collective suitability 

• We agree with the reviewer that collective suitability is more complex than just combining the suitability of individual crops (as also raised by reviewer 1). We have amended the manuscript by making this distinction clear throughout the text and also rewritten the discussion to more cautiousness with this respect. We have limited the results and discussions to indicate multiple and dual crop suitability only and not combined suitability or diversification.

Calculate in km² or he the areas both current and future scenarios 

• Done. Area calculations performed on the results and added in the results section. 

Use another approach. With current results it is not possible to state that the species can be cultivated.

• We acknowledge the reviewer’s concerns about a possibly limited view on suitability (as detailed above if soil or other components of the agricultural system are assumed as constant). As detailed above, we have amended the manuscript accordingly and detailed the limitations of our suitability definition which only considers climate, but no soil or other socio-ecological constraints. We have limited the results to multiple or dual suitability and not combined suitability or diversification.

---

## [Decision Letter · Decision Letter 1]

15 Jun 2020

Impacts of climate change on agro-climatic suitability of major food crops in Ghana

PONE-D-20-04571R1

Dear Dr. Chemura,

We’re pleased to inform you that your manuscript has been judged scientifically suitable for publication and will be formally accepted for publication once it meets all outstanding technical requirements.

Kind regards,

Shahid Farooq, Ph.D.

Academic Editor

PLOS ONE

Additional Editor Comments (optional):

I have received the reports from the referees. The comments of both reviewers were addressed by the authors. Therefore, the current version can be accepted for publication.

Reviewers' comments:

Reviewer's Responses to Questions

**Comments to the Author**

1. If the authors have adequately addressed your comments raised in a previous round of review and you feel that this manuscript is now acceptable for publication, you may indicate that here to bypass the “Comments to the Author” section, enter your conflict of interest statement in the “Confidential to Editor” section, and submit your "Accept" recommendation.

Reviewer #1: All comments have been addressed

Reviewer #2: All comments have been addressed

2. Is the manuscript technically sound, and do the data support the conclusions?

Reviewer #1: Yes

Reviewer #2: Yes

3. Has the statistical analysis been performed appropriately and rigorously? 

Reviewer #1: Yes

Reviewer #2: Yes

4. Have the authors made all data underlying the findings in their manuscript fully available?

Reviewer #1: Yes

Reviewer #2: Yes

5. Is the manuscript presented in an intelligible fashion and written in standard English?

Reviewer #1: Yes

Reviewer #2: Yes

6. Review Comments to the Author

Reviewer #1: (No Response)

Reviewer #2: I have evaluated the revised manuscript. The authors have addressed all the queries raised by during the review process. Therefore, I recommend accepting the manuscript in the current form.

7. PLOS authors have the option to publish the peer review history of their article (what does this mean?). If published, this will include your full peer review and any attached files.

Reviewer #1: No

Reviewer #2: No

---

## [Editor Report · Acceptance letter]

18 Jun 2020

PONE-D-20-04571R1 

Impacts of climate change on agro-climatic suitability of major food crops in Ghana 

Dear Dr. Chemura:

I'm pleased to inform you that your manuscript has been deemed suitable for publication in PLOS ONE. Congratulations! Your manuscript is now with our production department. 

Kind regards, 

on behalf of

Dr. Shahid Farooq 

Academic Editor

PLOS ONE